# Towards Understanding {10-11}-{10-12} Secondary Twinning Behaviors in AZ31 Magnesium Alloy during Fatigue Deformation

**DOI:** 10.3390/ma17071594

**Published:** 2024-03-31

**Authors:** Yunxiang You, Li Tan, Yuqin Yan, Tao Zhou, Pengfei Yang, Jian Tu, Zhiming Zhou

**Affiliations:** 1School of Materials Science and Engineering, Chongqing University of Technology, Chongqing 400054, China; youyunxiang@stu.cqut.edu.cn (Y.Y.);; 2Chongqing Yujiang Die-Casting Co., Ltd., Chongqing 400000, China; 3College of Engineering, Zhejiang Normal University, Jinhua 321004, China

**Keywords:** magnesium alloy, {10-11}-{10-12} secondary twins, local strain accommodation, fatigue deformation

## Abstract

Tensile-compression fatigue deformation tests were conducted on AZ31 magnesium alloy at room temperature. Electron backscatter diffraction (EBSD) scanning electron microscopy was used to scan the microstructure near the fatigue fracture surface. It was found that lamellar {10-11}-{10-12} secondary twins (STs) appeared inside primary {10-11} contraction twins (CTs), with a morphology similar to the previously discovered {10-12}-{10-12} STs. However, through detailed misorientation calibration, it was determined that this type of secondary twin is {10-11}-{10-12} ST. Through calculation and analysis, it was found that the matrix was under compressive stress in the normal direction (ND) during fatigue deformation, which was beneficial for the activation of primary {10-11} CTs. The local strain accommodation was evaluated based on the geometric compatibility parameter (m’) combined with the Schmid factor (SF) of the slip system, leading us to propose and discuss the possible formation mechanism of this secondary twin. The analysis results indicate that when the local strain caused by basal slip at the twin boundaries cannot be well transmitted, {10-11}-{10-12} STs are activated to coordinate the strain, and different loading directions lead to different formation mechanisms. Moreover, from the microstructure characterization near the entire fracture surface, we surmise that the presence of such secondary twins is not common.

## 1. Introduction

Magnesium alloys have characteristics such as low density, good damping properties, high specific stiffness, and good processability, which have attracted increasing attention from scholars, mainly for uses in aerospace, automotives, and 3C electronic products [1,2,3]. Among all reported twinning modes of magnesium alloys, {10-12} extension twinning is the most common activation mode, usually occurring when tensile stress is applied along the c-axis of the grain [4]. When the loading direction is parallel to the c-axis compression or perpendicular to the c-axis tension, {10-11} contraction twins and {10-11}-{10-12} STs will occur to coordinate deformation [5,6,7].

At present, some scholars have reported on {10-11}-{10-12} STs. Wu et al. [8] studied the twinning mode of the AZ31 magnesium alloy under uniaxial compression loading, and the results revealed that {10-11} contraction twinning and {10-11}-{10-12} secondary twinning are the main twinning modes during compression along the c-axis. When a uniaxial compressive stress is applied to the magnesium alloy along the normal direction (ND), dislocation pile-ups at the grain boundaries cause stress concentration, leading to the activation of primary {10-11} CTs, and as deformation continues, {10-11}-{10-12} STs activate and grow within the primary {10-11} CT to coordinate deformation. During tensile-compression fatigue deformation, traces of {10-11}-{10-12} STs can be found on the fatigue fracture surface, and some scholars have indicated that cyclic hardening phenomena will activate a large number of STs in the fatigue deformation, and the stress concentration at the boundaries of STs leads to crack initiation and ultimate fracture failure [9]. In a fully reversed strain-controlled fatigue test, Yu et al. [10] found that fatigue microcracks initiate at the {10-12} twin boundaries, but macroscopic crack propagation leading to final fracture occurs along the {10-11}-{10-12} secondary twin boundaries. The authors pointed out that the type of twin cannot be accurately determined solely by the deformation degree of the fracture surface, and it should only be judged based on the trace of the cleavage plane. Yang et al. [2] investigated the secondary twinning behavior during fatigue tests and proposed several local stress-induced secondary twinning mechanisms. The results showed that the interaction between the basal slip within primary twins and the twin boundaries plays an important role in the activation of secondary twins. Compared to slip–twin interaction, the local strain associated with twin–twin interaction may not directly induce the activation of such {10-12}-{10-12} STs. But, to date, our understanding of the induced mechanism of the {10-11}-{10-12} ST during fatigue tests is still incomplete.

In the previous study [11], we also found the presence of {10-11}-{10-12} STs in the fatigue fracture region, but as their quantity was relatively small, the activation mechanism of these twins was not revealed. Therefore, this paper analyzes the formation mechanism of {10-11}-{10-12} STs during fatigue deformation, elucidating this using the Schmid factor and geometric compatibility parameter. The findings provide insight into the activation of {10-11}-{10-12} STs with a low or negative SF in magnesium.

## 2. Materials and Methods

The stress–strain curves for the first cycle, the second cycle, and the half-life cycle were charted in the previous study, and the specific fatigue test parameters were also detailed [12]. The same as-rolled AZ31 Mg alloy sheet underwent fully reversed tension-compression fatigue tests at a total strain amplitude of 1% (fatigue testing machine model: MTS809, Metes Industrial Systems, Inc, Eden Prairie, MN, USA). One electro-polished with commercial AC2 solution, subsequently, the microstructure of the fatigue samples was characterized by scanning electron microscopy (SEM) and electron backscatter diffraction (EBSD) techniques. EBSD examination was performed using the HKL channel 5 system (Technology-Oxford Instruments, Abingdon, UK) in a scanning electron microscope (SEM, FEI Nova 400, Hillsboro, OR, USA).

## 3. Results

The hysteresis loops of the fatigue deformation process were discussed in the previous study [12]. This article only discusses the {10-11}-{10-12} STs that can be observed on the fatigue fracture surface. The IPF map in Figure 1a exhibits the microstructural features of the fracture surface after tensile-compression fatigue, with a specific region selected for subsequent analysis. The AZ31 sample shows a strong basal texture, as shown in Figure 1b. Table 1 lists the twinning types in magnesium alloy, the misorientation angles between twins and the matrix, and the corresponding rotation axes. The color legend for the IPF maps includes low-angle grain boundaries (2–15°), high-angle grain boundaries (>15°), and seven possible twinning boundaries [13], shown in different colors in Figure 2d. The six variants generate three types of misorientation, which are 60.01° <10-10>, 60.41° <8-1-70>, and 7.41° <1-210>, and the identification of the active twin variant is performed by analyzing the twin plane traces and misorientation [14]. 

Figure 2 shows the EBSD maps of the main analysis region. Figure 2b depicts an irregular lamellar structure composed of different colored boundaries. The red lines represent the twinning boundaries of {10-12} extension twins (ETs), and the yellow lines represent the twin boundaries of {10-11} CTs. There are two typical types of twins: one is the {10-11} CTs activated by the matrix, and the other type is the {10-11}-{10-12} STs activated within the {10-11} twins. And the {10-11} primary twin boundaries are composed of {10-12}, {10-11}, and {10-11}-{10-12} twin boundaries. In Figure 2c, a peak at around 15° can be observed, which may be caused by low-angle grain boundaries. The distribution peak at around 86° is associated with {10-12} ETs (86°/<1-210>), the peak near 56° is associated with {10-11} CTs (56°/<1-210>), and the peak near 38° is related to {10-11}-{10-12} STs (38°/<1-210>) [16].

The orientation relationship between the matrix and twins in Figure 2b is shown in Figure 3, where the matrix and twins are represented by different symbols. Pt1 (primary twin) is the primary {10-11} CTs activated by the matrix, while St1 (secondary twin) is the {10-11}-{10-12} STs activated within Pt1. From the (0001) and (10-12) pole figures, it can be observed that the CT variant V5 (Pt1) is activated in the matrix by compression along the ND, causing the orientation to tilt from ND to the rolling direction (RD). Subsequently, ET variant V3 (St1) is activated in Pt1 during the subsequent deformation process. Therefore, it can be identified that these lamellar structures are secondary twins. In this work, the identification of {10-11}-{10-12} STs is determined by the misorientation angle that occurs in the primary {10-11} CTs, as shown in Figure 4.

## 4. Discussion

### 4.1. Analyses of Schmid Factors

In order to reveal the activation mechanism of the observed {10-11}-{10-12} STs, the Schmid factors (SFs) of the primary {10-11} CTs, {10-12} ETs, and {10-11}-{10-12} STs are calculated. The Euler angle values are extracted from the EBSD results. Due to the rapid fracture of the specimen when cracks initiate [10], the final loading direction before fracture during fatigue deformation is uncertain. Therefore, two loading directions are considered in the calculation process: tension and compression along the ND. The twins in Figure 2a are represented by different numbers as shown in Figure 5a, and Figure 5b is a grain boundary map identified based on the axis and angle. The SF is defined as
*m* = cos*ϕ* × cos*λ*
where *λ* represents the angle between the loading axis and the normal twinning plane and *ϕ* stands for the angle between the loading axis and shear direction. When compressed along the ND, the loading direction is parallel to the c-axis, primary {10-11} CTs are activated when the loading direction is parallel to the c-axis, and {10-11}-{10-12} STs are activated after re-twinning within the primary twin. The SF values of the six potential {10-11} CT variants for the marked grains are shown in Table 2, and the SF values of the actual activated primary twin variants are highlighted in yellow in the table. The SF values of the six potential {10-11} CT variants for grains 1, 4, and 7 are all relatively high, indicating that compression along the ND is favorable for the activation of {10-11} CTs. Table 3 shows the SF values of the six potential {10-12} ET variants for the marked grains. From the data in the table, it can be observed that the SF values of both potential {10-12} ET variants of the matrix and the actual activated variants are negative, indicating that compression along the ND is unfavorable for activating {10-12} ETs within the matrix. The SF values of the actual activated {10-12} ET variants of primary twin 2 and twin 5, whether under compressive stress (−0.011, −0.008) or tensile stress (0.011, 0.008), do not comply with the Schmid law. Therefore, when the matrix is under compressive stress along the ND, {10-11} CTs 2, 5, and 8 are activated, with their SFs being 0.343, 0.304, and 0.451, respectively. As the SF values of all six variants are relatively high, the variants (0-111)[0-11-2] (V5) and (01-11)[01-1-2] (V6) are activated. Twins 2 and 5 activate the {10-11}-{10-12} ST variants (10-12)[−1011] (V4) during subsequent deformation processes, which possibly coordinate the local strain generated by the interaction between the twins or slip systems, which will be discussed further in the following paragraphs.

When tensile along the ND, the SF values of the potential six {10-11} CT variants of the marked grains are as shown in Table 4, and the SF values of the six potential {10-12} ET variants are as listed in Table 5. The SF values of the activated ST variants are highlighted in the table. The data in the table indicate that the SF values of the six potential {10-11} CT variants for the matrixes 1, 4, and 7 are negative, confirming that the {10-11} CTs are activated by compression along the ND. However, the SF values of these {10-11}-{10-12} ST variants are relatively low, displaying non-Schmid behavior. This indicates that compared with the external stress, the internal stress (or local effect) should play a more important role in the formation of the {10-11}-{10-12} STs.

For magnesium alloys, the contraction of the c-axis is mainly regulated by the formation of {10-11}-{10-12} STs, which will result in the reorientation of 37.5° and 30.1° twin/matrix [17]. Studies have shown that the SF usually controls the selection of variants for initial twin nucleation [15], and twin variants with higher SF values are more easily activated in plastic deformation [18]. However, at least half of the twin activation during certain deformation processes does not follow Schmid’s law, and even twin variants with lower or negative SF values can be activated [17,19].

Research has shown that when the shear strain caused by dislocation slip within a grain cannot be accommodated, twins within the same or neighboring grains can be activated to accommodate the local strain caused by dislocation slip or twinning [20,21,22]. Therefore, two possible mechanisms have been proposed for the formation of secondary twins with lower SF values: (i) when the strain caused by dislocation slip within the primary twin (PT) cannot be well transmitted to the matrix, the secondary twin (ST) is activated to accommodate the local strain; (ii) the strain induced by dislocation slip within the matrix at the primary twin boundary causes the formation of the secondary twin (ST). 

The schematic diagrams of these two possible mechanisms are shown in Figure 6. In order to evaluate the effectiveness of these mechanisms for the formation of a secondary twin, a geometric compatibility parameter (m’) is employed here and defined as m’ = cosφ × cosκ, where φ is the angle of the twin or slip plane normal and κ presents the angle between the shear directions of twin or dislocation slip. This factor was successively applied to elucidate the occurrence of the twin with a low SF or for variant selection of the paired twins in magnesium and titanium, where a relatively high m’ value indicates a better accommodation effect [2]. Therefore, m’ and SF values related to the twin variants and slip systems (the geometric relationship in the plane as shown in grain 1–6 in Figure 5a) associated with the two mechanisms mentioned above were calculated and will be discussed in more detail.

### 4.2. m’ Associated with Secondary Twin and Basal Slip within the Primary Twin

{10-11}-{10-12} STs can be activated within the primary {10-11} CTs, and a large number of dislocations pile up at the twin boundaries. Studies have shown that when the dislocation slip generated by the primary twin encounters a “hard” grain boundary, which occurs when the shear strain caused by the dislocation slip within the primary twin cannot be accommodated by its neighboring twin or matrix, the secondary twin can be activated within the same twin to accommodate the strain [21,22]. Therefore, taking secondary twins in the research region (Figure 5a) as an example, Table 6 lists the m’ values between the basal slip system of the primary twin and the secondary twin variants within twin 2, and Table 7 lists the m’ values between the basal slip system of the primary twin and the secondary twin variants within twin 5, with the corresponding SF values of the basal slip system within the primary twin also in Table 7. 

From the table data, it can be seen that when the loading direction is tensioned along the ND, the SF values of the basal slip systems (0001) [1-210] and (0001) [−1-120] in the primary twin are higher, and the accommodation between the basal slip system (0001) [−1-120] and the secondary twin variant is also better (m’ = 0.461). The m’ value between the basal slip system (0001)[−1-120] (the basal slip system with the maximum SF value) within the twin and the secondary ET variant is 0.461, as shown in Figure 7b, indicating that in this case, the secondary twins 3 and 6 are activated to accommodate the local strain incompatibility at the primary twin boundary caused by the basal slip. Similar to previous studies, the activation of secondary twins can reduce the strain incompatibility caused by the basal slip within the primary twins to the greatest extent [22]. Therefore, it can be inferred that the first type of proposed mechanism for the formation of secondary twins is effective under tensile loading.

### 4.3. m’ Associated with Secondary Twin and Basal Slip in the Matrix

The second mechanism of formation is the local strain-induced secondary twin (ST) at the primary twin boundary caused by the basal slip within the matrix. Previous studies have found that stress concentration induced by basal slip at the grain boundary within the matrix can induce twins, known as the matrix slip-induced twin mechanism [23,24,25]. From the data in Table 8, it can be seen that the SF values of the slip systems within matrix 1 are relatively low when the stress is applied along the ND, with the basal system (0001)[-12-10] being the highest at 0.107. In the process of plastic deformation, a low value of the Schmid factor typically indicates that the system is inactive in this loading direction [14,26]. Therefore, in this loading direction, the orientation of the primary twin is more favorable for the activation of basal slip systems than the original matrix orientation, as shown in Figure 4, so the SF of the basal slip systems within the primary twin is higher than that within the matrix. At the same time, it was found through calculations that when the compressive loading is along the ND, the accommodation between the basal slip systems within the primary twin and the secondary twin variants is unfavorable, with m’ values being zero or negative. The m’ values between the basal slip systems within the matrix and the six potential secondary twin variants are listed in Table 9. In this case, the basal slip systems (0001)[1-210] and (0001)[−1-120] in matrix 4 are more easily activated (under compressive loading in matrix 4: SF(0001)[−1-120] = 0.137, SF(0001)[1-210] = 0.139). And the m’ value between the slip system (0001)[−1-120] and the secondary twin variant 6 is 0.701, as shown in Figure 7a, which shows good accommodation. In conclusion, this study has found that the activation of most basal slip systems within the matrix is unfavorable, while basal slip systems within the matrix show good accommodation with the secondary twin variants when the compressive loading is along the ND. Therefore, it can be inferred that the second type of proposed mechanism for the formation of secondary twins is effective under compressive loading.

Finally, it should be pointed out that the change in loading direction can lead to different accommodation mechanisms. In this study, under tensile loading along the ND, the second accommodation mechanism is unfavorable, and the dislocation slip within the primary twin plays an important role in the activation of {10-11}-{10-12} STs. Meanwhile, under compressive loading along the ND, the first accommodation mechanism is unfavorable, and the dislocation slip within the matrix plays an important role in the activation of {10-11}-{10-12} STs. Through analysis, it is shown that although the SF values of the secondary twin variants are very low under the tensile-compression loading, the different loading direction determines the activation of different slip systems, which plays a crucial role in subsequent strain accommodation.

## 5. Conclusions

This paper has studied the AZ31 magnesium alloy through tension-compression fatigue tests at room temperature and provided EBSD characterization of the microstructure near the fatigue fracture, and the conclusions are as follows: (1)The lamellar secondary twin is activated inside the primary {10-11} CT. This morphological feature is similar to the previously discovered {10-12}-{10-12} STs. After misorientation calibration, it is found that this type of secondary twin is a {10-11}-{10-12} ST.(2)Local strain accommodation plays an important role in the formation of the unusual {10-11}-{10-12} ST. The calculation analysis using the Schmid factor and geometric compatibility parameter concludes that during tensile loading, when the strain caused by the basal slip within the {10-11} primary twin cannot be effectively transmitted to the matrix, it induces the {10-11}-{10-12} secondary twin within the primary twin to reduce the local strain incompatibility at the twin boundary. Under compressive loading, the activation of certain slip systems within the matrix induces local strain at the primary twin boundary, leading to the activation of {10-11}-{10-12} secondary twins.

## Figures and Tables

**Figure 1 materials-17-01594-f001:**
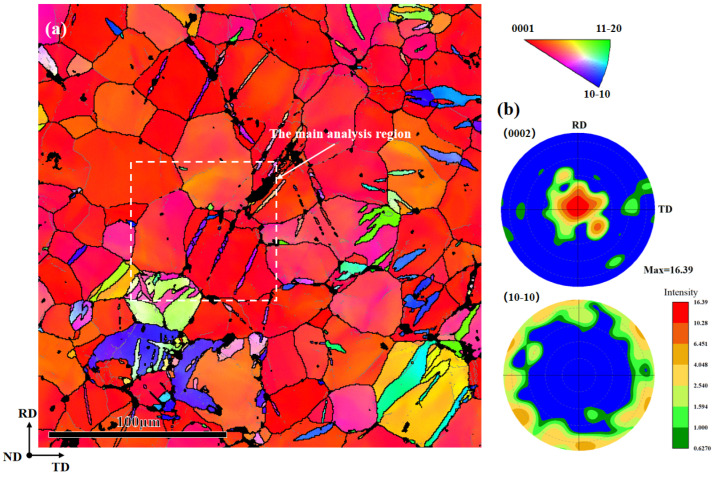
(**a**) Inverse pole figure map displaying the microstructural characteristics of the fatigued specimens used in the current work. (**b**) The corresponding pole figure showing a basal texture.

**Figure 2 materials-17-01594-f002:**
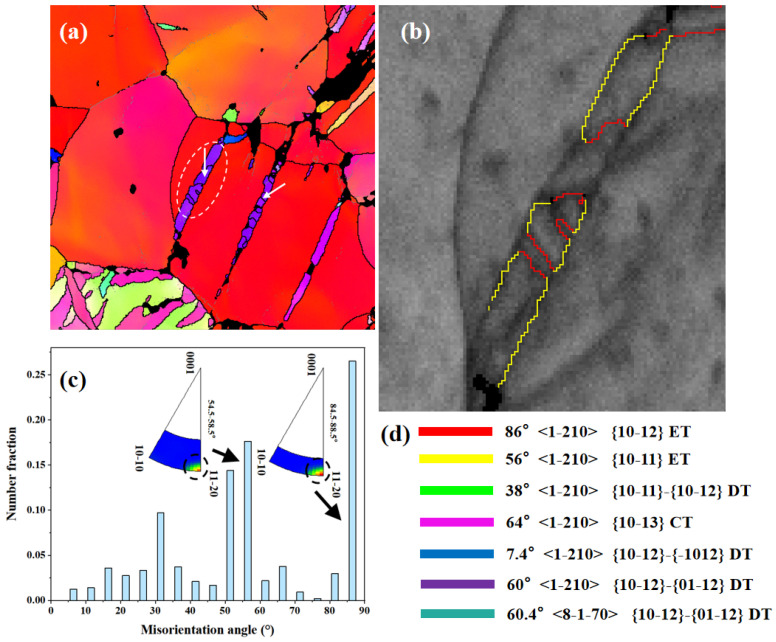
(**a**) IPF map of the main analysis region; (**b**) enlarged image of the region shown by the ellipse; (**c**) misorientation angle map corresponding to the region; (**d**) boundary color, axis, and twin type. The white arrows in the figure represent the direction of the point-to-point misorientation line profiles. The colorful lines represent the colors of different twin boundaries.

**Figure 3 materials-17-01594-f003:**
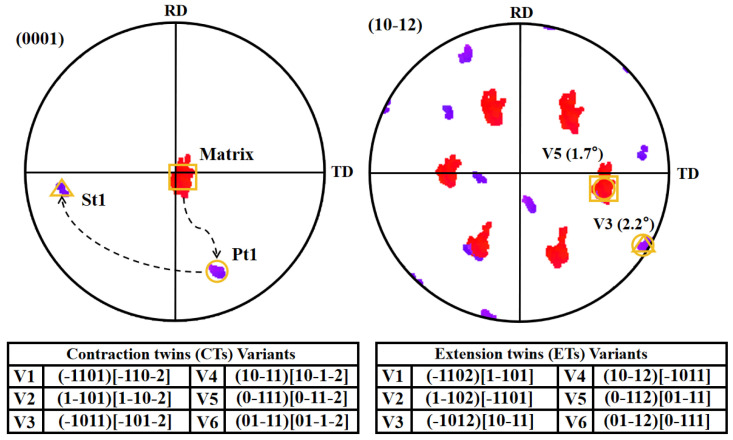
Corresponding orientation relationship of the tagged twin (Matrix, Pt1, St1) and the matrix in Figure 2b, respectively. The symbols indicate the experimentally projected sites of both twins and the matrix by EBSD.

**Figure 4 materials-17-01594-f004:**
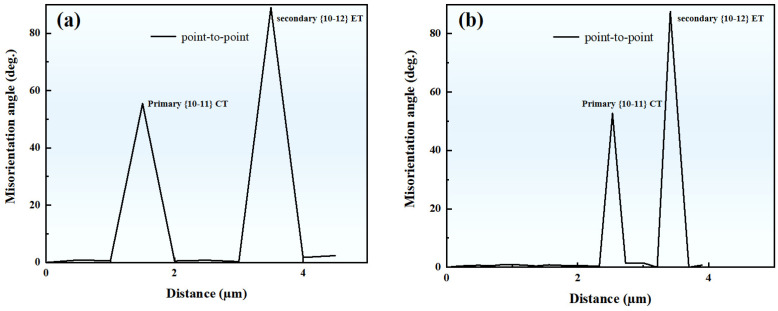
Point-to-point misorientation line profiles along the direction indicated by the white arrows in Figure 2a. (**a**) represents the white line on the left, and (**b**) represents the white line on the right.

**Figure 5 materials-17-01594-f005:**
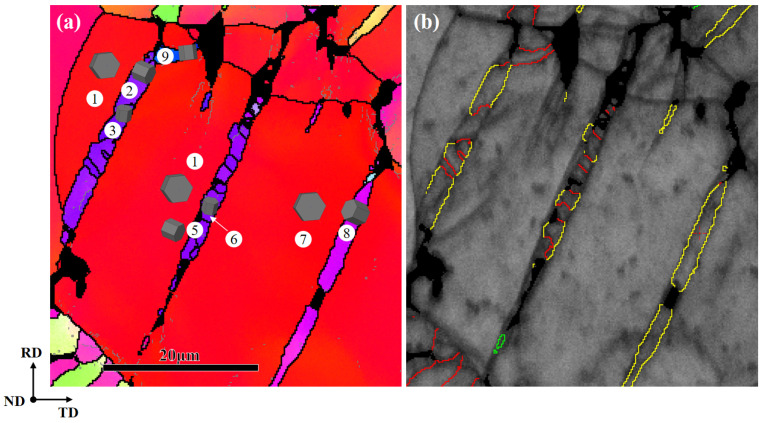
(**a**) IPF map indicating different twin variants with numerical labels; (**b**) grain boundary map.

**Figure 6 materials-17-01594-f006:**
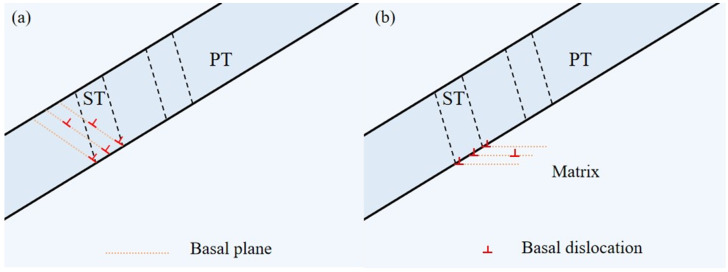
Schematics showing the possible formation mechanisms of a secondary twin within the primary twin. (**a**) The first mechanism above: Primary Twin (PT) basal slip-induced Secondary Twin (ST); (**b**) The second mechanism above: Matrix basal slip-induced Secondary Twin (ST).

**Figure 7 materials-17-01594-f007:**
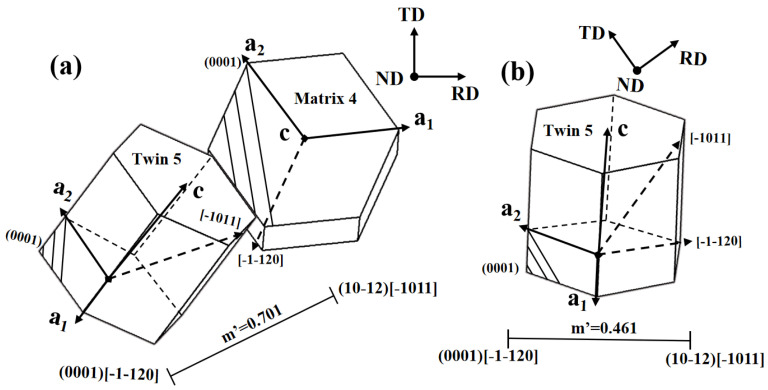
Two induced mechanism diagrams under different loading directions: (**a**) the compressive loading is along the ND direction; (**b**) the tensile loading is along the ND direction.

**Table 1 materials-17-01594-t001:** Common twin types and their angles and axes in magnesium (Mg) [15].

Type of Twin	Twin Plane	Misorientation Axis	Misorientation Angle
Extension twin	{10-12}	<1-210>	86.3°
Contraction twin	{10-11}	<1-210>	56°
	{10-13}	<1-210>	64°
Secondary twin	{10-11}-{10-12}	<1-210>	38°
	{10-13}-{10-12}	<1-210>	22°
	{10-12}-{−1012}	<1-210>	7.4°
	{10-12}-{01-12}	<1-210>	60°
	{10-12}-{0-112}	<8-1-70>	60.4°

**Table 2 materials-17-01594-t002:** SF values of the six potential {10-11} contraction twin variants labeled by number (compressive loading).

SF	(01-11)[01-1-2]	(10-11)[10-1-2]	(1-101)[1-10-2]	(0-111)[0-11-2]	(−1011)[-101-2]	(−1101)[-110-2]
1 (Matrix)	0.466	0.428	0.365	② 0.343	0.390	0.450
2 (PT)	−0.004	0.127	−0.124	−0.464	−0.088	0.121
3 (ST)	−0.057	0.095	−0.125	−0.481	−0.113	0.092
4 (Matrix)	0.482	0.445	0.356	⑤ 0.304	0.358	0.447
5 (PT)	0.013	0.135	−0.104	−0.458	−0.099	0.134
6 (ST)	0.013	0.133	−0.095	−0.458	−0.107	0.135
7 (Matrix)	⑧ 0.451	0.435	0.394	0.370	0.390	0.430
8	−0.343	−0.022	0.251	0.207	0.251	−0.027
9	−0.393	−0.054	−0.164	−0.421	−0.065	−0.146

The circled numbers represents the actual secondary twin variant activated by this parent/cube. The number indicates which variant of parent/cube is observed.

**Table 3 materials-17-01594-t003:** SF values of the six potential {10-12} extension twin variants labeled by number (compressive loading).

SF	(0-112)[01-11]	(−1012)[10-11]	(−1102)[1-101]	(01-12)[0-111]	(10-12)[−1011]	(1-102)[−1101]
1	−0.493	−0.494	−0.485	−0.479	−0.490	−0.495
2	0.255	−0.035	0.016	0.308	−0.011	−0.012
3	0.299	−0.001	0.032	0.348	0.023	0.007
4	−0.483	−0.488	−0.477	−0.463	−0.478	−0.488
5	0.240	−0.035	−0.005	0.294	−0.008	−0.032
6	0.240	−0.030	−0.011	0.294	−0.002	−0.037
7	−0.498	−0.498	−0.494	−0.489	−0.493	−0.498
8	0.113	−0.122	−0.151	0.050	−0.153	−0.119
9	0.488	0.071	0.187	0.491	0.072	0.185

**Table 4 materials-17-01594-t004:** SF values of the six potential {10-11} contraction twin variants labeled by number (tensile loading).

SF	(01-11)[01-1-2]	(10-11)[10-1-2]	(1-101)[1-10-2]	(0-111)[0-11-2]	(−1011)[−101-2]	(−1101)[−110-2]
1	−0.466	−0.428	−0.365	−0.343	−0.390	−0.450
2	0.004	−0.127	0.124	0.464	0.088	−0.121
3	0.057	−0.095	0.125	0.481	0.113	−0.092
4	−0.482	−0.445	−0.356	−0.304	−0.358	−0.447
5	−0.013	−0.135	0.104	0.458	0.099	−0.134
6	−0.013	−0.133	0.095	0.458	0.107	−0.135
7	−0.451	−0.435	−0.394	−0.370	−0.390	−0.430
8	0.343	0.022	−0.251	−0.207	−0.251	0.027
9	0.393	0.054	0.164	0.421	0.065	0.146

**Table 5 materials-17-01594-t005:** SF values of the six potential {10-12} extension twin variants labeled by number (tensile loading).

SF	(0-112)[01-11]	(−1012)[10-11]	(−1102)[1-101]	(01-12)[0-111]	(10-12)[−1011]	(1-102)[−1101]
1	0.493	0.494	0.485	0.479	0.490	0.495
2	−0.255	0.035	−0.016	−0.308	③ 0.011	0.012
3	−0.299	0.001	−0.032	−0.348	−0.023	−0.007
4	0.483	0.488	0.477	0.463	0.478	0.488
5	−0.240	0.035	0.005	−0.294	⑥ 0.008	0.032
6	−0.240	0.030	0.011	−0.294	0.002	0.037
7	0.498	0.498	0.494	0.489	0.493	0.498
8	−0.113	0.122	0.151	−0.050	0.153	0.119
9	−0.488	−0.071	−0.187	−0.491	−0.072	−0.185

The circled numbers represents the actual secondary twin variant activated by this parent/cube. The number indicates which variant of parent/cube is observed.

**Table 6 materials-17-01594-t006:** The m’ values of the basal slip system within the primary twin 2 and the secondary twin variant, and the corresponding SF values of the two kinds of systems.

M’	Secondary Twin Variant		(01-12)[0-111]	(10-12)[−1011]	(1-102)[−1101]	(0-112)[01-11]	(−1012)[10-11]	(−1102)[1-101]
basal slip system within primary twin	SF (tensile loading)	SF (compressive loading)	−0.308/0.308	0.011/−0.011	0.012/−0.012	−0.255/0.255	0.035/−0.035	−0.016/0.016
(0001)[2-1-10]	−0.015	0.015	0.000	−0.461	−0.461	0.000	0.461	0.461
(0001)[11-20]	−0.365	0.365	−0.461	−0.461	0.000	0.461	0.461	0.000
(0001)[1-210]	0.349	−0.349	0.461	0.000	−0.461	−0.461	0.000	0.461
(0001)[-12-10]	−0.349	0.349	−0.461	0.000	0.461	0.461	0.000	−0.461
(0001)[-1-120]	0.365	−0.365	0.461	0.461	0.000	−0.461	−0.461	0.000
(0001)[-2110]	0.015	−0.015	0.000	0.461	0.461	0.000	−0.461	−0.461

**Table 7 materials-17-01594-t007:** The m’ values of the basal slip system within the primary twin 5 and the secondary twin variant, and the corresponding SF values of the two kinds of systems.

m’	Secondary Twin Variant		(01-12)[0-111]	(10-12)[−1011]	(1-102)[−1101]	(0-112)[01-11]	(−1012)[10-11]	(−1102)[1-101]
basal slip system within primary twin	SF (tensile loading)	SF (compressive loading)	0.295/−0.295	−0.002/0.002	−0.037/0.037	0.24/−0.24	−0.03/0.03	−0.011/0.011
(0001)[2-1-10]	−0.005	0.005	0.000	−0.461	−0.461	0.000	0.461	0.461
(0001)[11-20]	−0.368	0.368	−0.461	−0.461	0.000	0.461	0.461	0.000
(0001)[1-210]	0.363	−0.363	0.461	0.000	−0.461	−0.461	0.000	0.461
(0001)[-12-10]	−0.363	0.363	−0.461	0.000	0.461	0.461	0.000	−0.461
(0001)[-1-120]	0.368	−0.368	0.461	0.461	0.000	−0.461	−0.461	0.000
(0001)[-2110]	0.005	−0.005	0.000	0.461	0.461	0.000	−0.461	−0.461

**Table 8 materials-17-01594-t008:** The m’ values of the basal slip system within matrix 1 and the secondary twin variant within twin 2, and the corresponding SF values of the two kinds of systems.

m’	Secondary Twin Variant		(01-12)[0-111]	(10-12)[−1011]	(1-102)[−1101]	(0-112)[01-11]	(−1012)[10-11]	(−1102)[1-101]
basal slip system within matrix	SF (tensile loading)	SF (compressive loading)	−0.308/0.308	0.011/−0.011	0.012/−0.012	−0.255/0.255	0.036/−0.036	−0.016/0.016
(0001)[2-1-10]	−0.024	0.024	−0.037	−0.456	−0.079	0.000	0.080	0.418
(0001)[11-20]	0.083	−0.083	−0.842	−0.674	−0.076	0.022	−0.003	−0.269
(0001)[1-210]	−0.107	0.107	0.805	0.218	−0.004	−0.021	0.083	0.687
(0001)[-12-10]	0.107	−0.107	−0.805	−0.218	0.004	0.021	−0.083	−0.687
(0001)[-1-120]	−0.083	0.083	0.842	0.674	0.076	−0.022	0.003	0.269
(0001)[-2110]	0.024	−0.024	0.037	0.456	0.079	0.000	−0.080	−0.418

**Table 9 materials-17-01594-t009:** The m’ values of the basal slip system within matrix 4 and the secondary twin variant within twin 5, and the corresponding SF values of the two kinds of systems.

m’	Secondary Twin Variant		(01-12)[0-111]	(10-12)[−1011]	(1-102)[−1101]	(0-112)[01-11]	(−1012)[10-11]	(−1102)[1-101]
basal slip system within matrix	SF (tensile loading)	SF (compressive loading)	−0.295/0.295	0.002/−0.002	0.037/−0.037	−0.24/0.24	0.03/−0.03	0.011/−0.011
(0001)[2-1-10]	−0.002	0.002	−0.012	−0.460	−0.115	0.000	0.114	0.448
(0001)[11-20]	0.137	−0.137	−0.852	−0.701	−0.105	0.007	0.007	−0.256
(0001)[1-210]	−0.139	0.139	0.840	0.240	−0.009	−0.007	0.108	0.705
(0001)[-12-10]	0.139	0.139	−0.840	−0.240	0.009	0.007	−0.108	−0.705
(0001)[-1-120]	−0.137	0.137	0.852	0.701	0.105	−0.007	−0.007	0.256
(0001)[-2110]	0.002	−0.002	0.012	0.460	0.115	0.000	−0.114	−0.448

## Data Availability

Data are contained within the article.

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
