# Peer review of "Towards Understanding {10-11}-{10-12} Secondary Twinning Behaviors in AZ31 Magnesium Alloy during Fatigue Deformation"

_materials, 2024, doi:10.3390/ma17071594_

Round 1
Reviewer 1 Report
Comments and Suggestions for Authors
The manuscript under consideration is “Towards understanding {10-11}-{10-12} secondary twinning behaviors in AZ31 Magnesium Alloy during fatigue deformation”. The authors demonstrate in-depth investigation of material near the fatigue fracture surface of AZ31 Magnesium Alloy. The peculiarities of twin formation have been established.
There are several minor remarks.
1. The manuscript is indicated as “Communication” but in line 1 “Article” is indicated.
2. Practical significance of research should be outlines in the Abstract.
3. Do magnesium alloys really possess high strength as it is stated in line 32?
4. “have been” repeated twice in line 90.
Comments on the Quality of English LanguageMinor editing of English language required
Author Response
Detailed Response to Reviewer 1#:
We truly appreciate your professional and informative comments on the manuscript. This manuscript has been carefully revised according to each comment. The following is the detailed response to your comments and suggestions. For clearance, we have listed the comments below and have addressed them one by one.
Comment 1. The manuscript is indicated as “Communication” but in line 1 “Article” is indicated.
Response: Thank you for finding this issue, the author has changed the type of the article to Communication. The revised parts have been highlighted by red color in the marked-up manuscript.
Comment 2. Practical significance of research should be outlines in the Abstract.
Response: Thanks for your constructive suggestion. The author has already written the significance of the study in the last paragraph of the Introduction (lines 68-71). However, due to the author’s oversight, the practical significance was not clearly stated. We apologize for this mistake. Based on your suggestion. The practical significance of the research is:
Fatigue damage is a common problem encountered in the service process of components and also one of the main issues restricting the application of alloys in various fields [1,2]. The {10-11}-{10-12} twin boundaries often serve as nucleation sites for microcracks in fatigue deformation [3]. Therefore, this work aims to understand the formation mechanism of {10-11}-{10-12} secondary twins in magnesium alloys through Schmid factor and geometric compatibility parameter, providing new insights for improving the fatigue performance of magnesium alloys.
- Yadav, V.K.; Gaur, V.; Singh, I.V. Effect of Pre-Corrosion on Crack Initiation Behavior of Friction Stir Welded Aluminum Alloy 2024-T3 in High and Low Cycle Fatigue Regimes. Int. J. Fatigue 2024, 179.
- Wang, C.; Zhang, T.; Zhang, T.; He, Y.; Bi, Y. Fatigue Performance and Alternate Coupling Mechanism of ZL114A Aluminum Alloy under Corrosion-Fatigue Alternating. Int. J. Fatigue 2024, 183, 108218.
- Chaudry, U.M.; Tariq, H.M.R.; Zubair, M.; Ansari, N.; Jun, T.-S. Implications of Twinning on the Microstructure Development, Crystallographic Texture and Mechanical Performance of Mg Alloys- a Critical Review. J. Magnes. Alloys 2023, 11, 4146–4165.
Comment 3. Do magnesium alloys really possess high strength as it is stated in line 32?
Response: Thanks for your constructive suggestion. This was an oversight by the author, who used a controversial term. The author has already removed the term ‘high strength’.
Comment 4. “have been” repeated twice in line 90.
Response: Thanks for you carefully review. The author sincerely apologizes for making a fundamental mistake, and has deleted the unnecessary ‘have been’. The revised parts have been highlighted by red color in the marked-up manuscript.
With best regards from all of the authors,
Sincerely Yours

Reviewer 2 Report
Comments and Suggestions for Authors
General remarks:
1. First of all, I would like to congratulate the authors for the quality of their work. The work is interesting, although there are several similar works that study aspects close to this work. The originality lies in focusing on the formation mechanism of {10-11}-{10-12} STs during fatigue deformation, elucidating the
formation mechanism of these STs using the Schmid factor and geometric compatibility parameter. It is highlighted that these STs are activated to adapt to a local stress, which is very important for understanding the alloy's response to cyclic loading. The objectives have been covered and the paper reads well and is adequately structured. In the opinion of the reviewer, the paper presented is mostly well written and interesting, but it must overcome some drawbacks.
2. The title and the intentions declared in the abstract correspond to the contents of the paper. The paper contains an abstract and an introduction which is in fact a critical review of the state of the art. Some of the references are old (12 references out of 27 are older than 10 years).
3. The authors have some important contributions in the subject of the present paper:
1. Yang, P.; Yang, Z.; Li, L.; Sun, Q.; Tan, L.; Ma, X.; Zhu, M. Towards understanding double extension twinning behaviors in magnesium alloy during uniaxial tension deformation. Journal of Alloys and Compounds 2022, 894, doi:10.1016/j.jallcom.2021.162491.
2. Tan, L.; Huang, X.; Wang, Y.; Sun, Q.; Zhang, Y.; Tu, J.; Zhou, Z. Activation Behavior of {10-12}-{10-12} Secondary Twins by Different Strain Variables and Different Loading Directions during Fatigue Deformation of AZ31 Magnesium Alloy. Metals 2022, 12, doi:10.3390/met12091433.
3. Li, L.; Yang, J.; Yang, Z.; Sun, Q.; Tan, L.; Zeng, Q.; Zhu, M. Towards revealing the relationship between deformation twin and fatigue crack initiation in a rolled magnesium alloy. Materials Characterization 2021, 179, doi:10.1016/j.matchar.2021.111362.
4. Bai, J.; Yang, P.; Yang, Z.; Sun, Q.; Tan, L. Towards understanding relationships between tension property and twinning boundaries in magnesium alloy. Metals 2021, 11, doi:10.3390/met11050745.
5. Tan, L.; Zhang, X.; Xia, T.; Huang, G.; Liu, Q. Relationship Between Secondary Twins and Pyramidal Dislocations in a Mg-3Al-1Zn Alloy During High Cycle Fatigue Deformation. Xiyou Jinshu Cailiao Yu Gongcheng/Rare Metal Materials and Engineering 2019, 48, 1435-1439.
6. Sun, Q.; Xia, T.; Tan, L.; Tu, J.; Zhang, M.; Zhu, M.; Zhang, X. Influence of {101¯2} twin characteristics on detwinning in Mg-3Al-1Zn alloy. Materials Science and Engineering: A 2018, 735, 243-249, doi:10.1016/j.msea.2018.08.051.
7. Shu, Y.; Zhang, X.Y.; Yu, J.P.; Tan, L.; Yin, R.S.; Liu, Q. Tensile behaviors of fatigued AZ31 magnesium alloy. Transactions of Nonferrous Metals Society of China (English Edition) 2018, 28, 896-901, doi:10.1016/S1003-6326(18)64723-5.
Few minor remarks:
4. Where is the novelty in this study with respect to other papers? What is their contribution, with this research work, to the academic-scientific community of sciences and engineering?
5. At the end of the Introduction chapter the authors must mention which is the novelty of the paper with respect to the papers presented in the state of the art. Which are the strong points of the present paper? What does the paper bring new related to other papers?
6. I want to mention that all references must be in accordance with the main topic of the research work, methods, results, and discussions; please avoid alteration of citations leading to bad scientific practices favouring authors or journals (there are 4 references of the authors of the present paper in the reference list!!!).
7. All the experiments are conducted at room temperature. Usually, the AZ31 magnesium alloy is used at higher temperatures. Do the authors consider that studying the alloy in other temperature conditions could be necessary?
8. How do STs in AZ31 compare to those in other magnesium alloys? Do the authors consider that a comparative study would prove useful?
9. On page 3, line 90, there are some repeating words (have been) and at the end of the sentence there is a comma instead of a dot. On page 7, line 183, there are some spaces between words. On page 8, line 195, there are two tabs at the beginning of the sentence. On page 8, line 222, there is a sentence that is found alone and without connection with other sentences (probably the title of a subchapter).
Author Response
Detailed Response to Reviewer 2#:
We truly appreciate your professional and informative comments on the manuscript. This manuscript has been carefully revised according to each comment. The following is the detailed response to your comments and suggestions. For clearance, we have listed the comments below and have addressed them one by one.
Comment 1. First of all, I would like to congratulate the authors for the quality of their work. The work is interesting, although there are several similar works that study aspects close to this work. The originality lies in focusing on the formation mechanism of {10-11}-{10-12} STs during fatigue deformation, elucidating the formation mechanism of these STs using the Schmid factor and geometric compatibility parameter. It is highlighted that these STs are activated to adapt to a local stress, which is very important for understanding the alloy's response to cyclic loading. The objectives have been covered and the paper reads well and is adequately structured. In the opinion of the reviewer, the paper presented is mostly well written and interesting, but it must overcome some drawbacks.
Response: Thank you for your recognition of this work. We will make corrections based on the suggestions and questions you have raised
Comment 2. The title and the intentions declared in the abstract correspond to the contents of the paper. The paper contains an abstract and an introduction which is in fact a critical review of the state of the art. Some of the references are old (12 references out of 27 are older than 10 years).
Response: Thank you for your recognition of this work. Based on your suggestion, we have added or edited citations of more recent references. The revised parts have been highlighted by red color in the marked-up manuscript.
Comment 4. Where is the novelty in this study with respect to other papers? What is their contribution, with this research work, to the academic-scientific community of sciences and engineering?
Response: Thank you for your recognition of this work. These studies form the basis of the research in this work. The fatigue damage is a common problem encountered in the service process of components and also one of the main issues restricting the application of alloys in various fields [1,2]. The {10-11}-{10-12} twin boundaries often serve as nucleation sites for microcracks in fatigue deformation [3]. Therefore, this work aims to understand the formation mechanism of {10-11}-{10-12} secondary twins in magnesium alloys through Schmid factor and geometric compatibility parameter, providing new insights for improving the fatigue performance of magnesium alloys.
- Yadav, V.K.; Gaur, V.; Singh, I.V. Effect of Pre-Corrosion on Crack Initiation Behavior of Friction Stir Welded Aluminum Alloy 2024-T3 in High and Low Cycle Fatigue Regimes. Int. J. Fatigue 2024, 179.
- Wang, C.; Zhang, T.; Zhang, T.; He, Y.; Bi, Y. Fatigue Performance and Alternate Coupling Mechanism of ZL114A Aluminum Alloy under Corrosion-Fatigue Alternating. Int. J. Fatigue 2024, 183, 108218.
- Chaudry, U.M.; Tariq, H.M.R.; Zubair, M.; Ansari, N.; Jun, T.-S. Implications of Twinning on the Microstructure Development, Crystallographic Texture and Mechanical Performance of Mg Alloys- a Critical Review. J. Magnes. Alloys 2023, 11, 4146–4165.
Comment 5. At the end of the Introduction chapter the authors must mention which is the novelty of the paper with respect to the papers presented in the state of the art. Which are the strong points of the present paper? What does the paper bring new related to other papers?
Response: The authors agree with the reviewer’s comment. In previous fatigue studies under tensile-compressive loading, {10-11}-{10-12} secondary twins (ST) were also observed. However, the mechanism of their formation is not yet clear. Through calculations, it has been found that the activation of these secondary twins does not follow the Schmid law under macroscopic loading conditions. Therefore, this work introduces geometric compatibility parameter to reveal the formation mechanism of this type of secondary twins in fatigue deformation from the perspective of crystallography. These finds may provide a new insight for improving the fatigue performance of magnesium alloys.
Comment 6. I want to mention that all references must be in accordance with the main topic of the research work, methods, results, and discussions; please avoid alteration of citations leading to bad scientific practices favouring authors or journals (there are 4 references of the authors of the present paper in the reference list!!!).
Response: These references from the authors themselves provide the necessary research foundation for this work. Upon your advice, we have made modifications to one of these references (the thirteenth). The revised parts have been highlighted by red color in the marked-up manuscript.
Comment 7 . All the experiments are conducted at room temperature. Usually, the AZ31 magnesium alloy is used at higher temperatures. Do the authors consider that studying the alloy in other temperature conditions could be necessary?
Response: Thanks for remind us this important points. The research in this work is all conducted under room temperature conditions. This is because, for practical applications, room temperature deformation can to some extent save production costs. However, the authors believe that research under other temperature conditions is also necessary. As you mentioned, magnesium alloys exhibit better plasticity at elevated temperatures due to the activation of prismatic <c+a> slip [4]. We would do the research in the future.
- 4. Chaudry, U.M.; Hamad, K.; Kim, J.-G. On the Ductility of Magnesium Based Materials: A Mini Review. J. Alloys Compd. 2019, 792, 652–
Comment 8.. How do STs in AZ31 compare to those in other magnesium alloys? Do the authors consider that a comparative study would prove useful?
Response: Thank you for raising a good question. From the authors perspective, twinning in magnesium and its alloys is a continuous process involving nucleation, propagation, and growth, which is related to the motion of twin boundaries (TB). The formation of secondary twins is mostly to accommodate compression or tension along the c-axis [5,6]. The grains in different types of magnesium alloys still exhibit a hexagonal close-packed (HCP) structure, so the formation mechanism of secondary twins (ST) and effects are basically consistent. At the same time, you may notice that there is currently almost no literature comparing twinning behavior between different types of magnesium alloys. In general, the authors are also looking forward to research in this area.
- Malik, A.; Wang, Y.; Nazeer, F. The Development of a Strong and Ductile Mg–Zn–Zr Thin Sheet through Nano Precipitates and Pre-Induced Dislocation. Mater. Sci. Eng. A 2021, 817, 141339.
- Malik, A.; Wang, Y.; Huanwu, C.; Nazeer, F.; Ahmed, B.; Khan, M.A.; Mingjun, W. Constitutive Analysis, Twinning, Recrystallization, and Crack in Fine-Grained ZK61 Mg Alloy during High Strain Rate Compression over a Wide Range of Temperatures. Mater. Sci. Eng. A 2020, 771, 138649.
Comment 9.. On page 3, line 90, there are some repeating words (have been) and at the end of the sentence there is a comma instead of a dot. On page 7, line 183, there are some spaces between words. On page 8, line 195, there are two tabs at the beginning of the sentence. On page 8, line 222, there is a sentence that is found alone and without connection with other sentences (probably the title of a subchapter).
Response: Thank you for reminding us. The mentioned issues have been corrected one by one, and the revised parts have been highlighted by red color in the marked-up manuscript.
With best regards from all of the authors,
Sincerely Yours
Reviewer 3 Report
Comments and Suggestions for Authors
The paper investigates the mechanism of secondary twinning formation in AZ31 Mg alloy during cyclic tension-compression deformation. The presented data is not very extensive, but it has been analyzed in detail and rather clearly illustrates the discussed issue / phenomenon. I suggest only to: (1) describe more in detail the parameters of EBSD scans and analysis; (2) re-order figures 3 and 4 to be more consistent with the way that they are mentioned in the text and (3) check the numbering of figures (e.g. Figure 4 in line 141).
Comments on the Quality of English LanguageThe minor editing of English language by the MDPI editorial office at the proof stage should be sufficient.
Some issues detected are mentioned below:
Line 74: "... and the specific fatigue test parameters were also detailed."
Line 90: "... fatigue deformation process have been have been discussed ..."
Line 281: "In conclusion, this study found that the activation of most basal slip systems within the matrix is inhibited, while basal slip systems within the matrix show good cooridation with the secondary twin variants when the compressive loading is along ND."
Author Response
Detailed Response to Reviewer 3#:
We truly appreciate your professional and informative comments on the manuscript. This manuscript has been carefully revised according to each comment. The following is the detailed response to your comments and suggestions. For clearance, we have listed the comments below and have addressed them one by one.
Comment 1. describe more in detail the parameters of EBSD scans and analysis
Response: Thanks for your good suggestion. Due to the word count limit of the article, the authors did not provide more details on EBSD. Specific parameters of EBSD are: acceleration voltage of 20 kV, sample tilt of 70°, acquisition speed of 1095 Hz, and step size of 0.5 μm.
Comment 2. re-order figures 3 and 4 to be more consistent with the way that they are mentioned in the text
Response: Thanks for your suggestion. The author has re-order Figures 3 and 4, which now align more with the logical flow of the article. The revised parts have been highlighted by red color in the marked-up manuscript.
Comment 3. check the numbering of figures (e.g. Figure 4 in line 141).
Response: Thank you for reminding us. There is a new numbering in the article, and the numbering of the current image has been corrected (e.g. Figure 4 in line 141). The revised parts have been highlighted by red color in the marked-up manuscript.
With best regards from all of the authors,
Sincerely Yours